# A Novel Isogenic Human Cell-Based System for MEN1 Syndrome Generated by CRISPR/Cas9 Genome Editing

**DOI:** 10.3390/ijms222112054

**Published:** 2021-11-08

**Authors:** Natalia Klementieva, Daria Goliusova, Julia Krupinova, Vladislav Yanvarev, Alexandra Panova, Natalia Mokrysheva, Sergey L. Kiselev

**Affiliations:** 1Endocrinology Research Centre, 115478 Moscow, Russia; j.krupinova@gmail.com (J.K.); a.v.panova@mail.ru (A.P.); nm70@mail.ru (N.M.); 2Vavilov Institute of General Genetics, Russian Academy of Sciences, 119991 Moscow, Russia; daria.goliusova@mail.ru (D.G.); dodmod@mail.ru (V.Y.)

**Keywords:** MEN1, induced pluripotent stem cells, CRISPR/Cas9 genome editing, isogenic cell lines, definitive endoderm differentiation

## Abstract

Multiple endocrine neoplasia type 1 (MEN1) is a rare tumor syndrome that manifests differently among various patients. Despite the mutations in the *MEN1* gene that commonly predispose tumor development, there are no obvious phenotype–genotype correlations. The existing animal and in vitro models do not allow for studies of the molecular genetics of the disease in a human-specific context. We aimed to create a new human cell-based model, which would consider the variability in genetic or environmental factors that cause the complexity of MEN1 syndrome. Here, we generated patient-specific induced pluripotent stem cell lines carrying the mutation c.1252G>T, D418Y in the *MEN1* gene. To reduce the genetically determined variability of the existing cellular models, we created an isogenic cell system by modifying the target allele through CRISPR/Cas9 editing with great specificity and efficiency. The high potential of these cell lines to differentiate into the endodermal lineage in defined conditions ensures the next steps in the development of more specialized cells that are commonly affected in MEN1 patients, such as parathyroid or pancreatic islet cells. We anticipate that this isogenic system will be broadly useful to comprehensively study *MEN1* gene function across different contexts, including in vitro modeling of MEN1 syndrome.

## 1. Introduction

Multiple endocrine neoplasia type 1 (MEN1) syndrome is a rare autosomal dominant inherited disorder that manifests as a variety of endocrine and non-endocrine tumors. Most frequently, combined adenomas of the parathyroid glands, neuroendocrine pancreas, and anterior pituitary gland are observed. This syndrome can affect people of all ages and still leads to decreased life expectancy, despite progress in medical care [1]. Germline heterozygous mutations of the *MEN1* gene are found in more than 90% of familial cases, wherein tumors in MEN1 patients usually show loss of heterozygosity (LOH). De novo *MEN1* mutations are revealed, albeit rarely, in sporadic tumors. However, an absence of phenotype–genotype correlations has been reported to date, since individuals with the same mutation may have different symptom manifestations, including a wide range of intrafamilial clinical heterogeneity [2,3]. There have been no reports on germline homozygous mutations presumably leading to embryonic death, which points to an important function of *MEN1* in early development [4].

The *MEN1* gene consists of 10 exons and encodes a 610-amino-acid protein, menin, which participates in transcription regulation, genome stability, and cell division and proliferation by interactions with a number of different proteins. Menin is highly conserved among species, from drosophila to humans [5]. Although a great deal of research has been carried out regarding the protein and gene partners of menin in cell and mouse models, along with an exploration of the mutation landscape in patients, there is still only a poor understanding of the molecular mechanisms of MEN1 syndrome development [6].

Currently, rapidly expanding technologies of cellular reprogramming and lineage differentiation offer a new possibility to investigate genome-based phenotypes in a dish [7]. Patient-specific induced pluripotent stem cells (iPSCs) have become a particularly advantageous tool for the exploration of disease-associated molecular pathways, affected by various factors from the genetic background to environmental conditions [8]. The differentiation abilities of iPSCs provide us with an opportunity to generate various cell lineages of the same genetic background, thus establishing an isogenic system to investigate not only the functions of the particular gene but also the epigenetic changes that can occur due to mutations and underline responses to metabolic or environmental signals. Evidence from protein–protein interaction studies has shown that menin is involved in epigenetic regulation and gene transcription control, affecting the expression of target genes [9,10]. Despite the increasingly widespread use of iPSC-based disease modeling, genetic variability among iPSC lines may conceal the effects of the causative mutation. Recent advancements in CRISPR/Cas9 genome editing open up new perspectives in gene modification, with high efficiency and accuracy, to meet this challenge [11,12]. In combination with iPSCs’ differentiation potential, genome editing enables the generation of isogenic cell lines of various lineages to investigate complex disorders such as MEN1.

Here, we first report an isogenic system based on iPSCs derived from a patient with familial MEN1 syndrome. We demonstrate the high potential of these cells to differentiate into the endodermal lineage as an essential step in the development of endocrine tissues. The generated patient-specific and CRISPR/Cas9-edited wild-type cell lines are a novel model suitable for studying the molecular mechanisms of MEN1 syndrome in vitro.

## 2. Results

### 2.1. Generation of a New iPSC Line from a Patient with MEN1 Syndrome

A dermal fibroblast primary cell culture was established from a skin biopsy of a 39-year-old female patient with familial MEN1 syndrome. The patient carries a missense heterozygous mutation (c.1252G>T) in exon 9 of the *MEN1* gene that was revealed by earlier genomic DNA sequence analysis [13]. To reprogram the patient’s fibroblasts into pluripotent cells, we chose a non-integrating RNA-based system carrying the puromycin resistance gene that provides great speed and efficiency of colony emergence with complete absence of integration and a very low aneuploidy rate [14]. Prior to transfection, an optimal puromycin concentration was determined by serial dilutions. The optimal concentration when 80% of fibroblasts died on Day 5 was 3 μg/mL. Under such conditions, fibroblasts were transfected with an RNA vector encoding five reprogramming factors—OCT4, KLF-4, SOX2, GLIS1, and c-MYC—and selected with puromycin. Approximately 3 weeks after transfection, dozens of colonies with human embryonic stem cell (ESC)-like morphology were observed. At this stage, the reprogramming efficiency was roughly estimated as 0.15%, determined by colony count. Colonies were manually picked for expansion and characterization. iPSCs exhibited a typical ESC morphology, with round, closely packed cells, a high nucleus-to-cytoplasm ratio, and prominent nucleoli (Figure 1A). The expression of pluripotency markers including nuclear transcription factor OCT4 and surface antigen SSEA-4 was observed in the established iPSCs (Figure 1B), as well as in the control ESCs (Appendix A).

Functional pluripotency was assessed by spontaneous differentiation into all three germ layers. Differentiation was performed in vitro through embryoid body formation. Immunohistochemical analysis revealed that patient-derived iPSCs were capable of differentiating into endodermal (Alpha-fetoprotein), mesodermal (Vimentin), and ectodermal (Beta-III-tubulin) lineage cells (Figure 1C). The genomic stability of the generated iPSC lines was examined by G-band analysis, which revealed a normal 46 XX karyotype (Appendix A). We also confirmed the presence of the disease-causing mutation (c.1252G>T) in the iPSCs by Sanger sequencing (Figure 1D).

In addition, we analyzed the expression of menin protein in the established patient-specific iPSC line, as well as in normal ESC line H9, using immunocytochemistry. Protein expression with localization mainly in the nucleus was observed in both cell lines (Figure 1E).

The completely characterized new iPSC line, named hiPSMEN1-MNA-6, derived from the patient carrying a mutation in the *MEN1* gene, was chosen for further experiments.

### 2.2. Capacity of the Generated iPSCs for Differentiation into Endodermal Cells

The modeling of the functional phenotype of a disease with human iPSCs involves their differentiation into the desired cell types. In the case of MEN1 syndrome, pancreatic and parathyroid endocrine cells, which are known to originate from the definitive endoderm (DE) during embryonic development, are of particular interest. Since individual human iPSC lines that are morphologically indistinguishable can be quite different in their differentiation potential, mostly due to the variety in the genetic backgrounds of donors [15,16], there is a need to evaluate the differentiation capacity of each established cell line.

Herein, we compared the differentiation capacities of the patient-derived hiPSMEN1-MNA-6 cell line and the human ESC line H9, commonly used as a benchmark in iPSC research, especially in comparative studies [17,18]. We differentiated both lines into DE cells, which are capable of further differentiation into more specialized cells of parathyroid or pancreatic lineage. Cells were passaged in parallel and then induced to differentiate for 5 days using the STEMdiff™ Definitive Endoderm Kit. We previously addressed initial cell-seeding density optimization to reduce cell death due to overgrowth during differentiation. The optimal cell density was 7.5 × 10^5^ cells/cm^2^, which was threefold lower than recommended. In such conditions, differentiating cultures reached 100% confluence on Day 3, with a minimal number of dead cells. On Day 5, the differentiated cells were fixed and analyzed for the expression of DE markers, including transcription factors SOX17 and FOXA2 and surface antigen CXCR4. The immunofluorescence staining showed a rather homogeneous expression of DE markers throughout the cell cultures derived from both hiPSMEN1-MNA-6 and H9 cells (Figure 2A), whereas no specific staining was observed in undifferentiated control iPSCs (Appendix A).

To more accurately compare the expression levels of the DE genes in these derivatives, quantitative flow cytometry assay was carried out. As expected, we observed large distinct populations of SOX17+, CXCR4+, and FOXA2+ cells in each cell line. To obtain a percentage of positive cells, the gate was adjusted to the autofluorescence of unstained cells. The purity of DE cells was in the range of 77–99% (Figure 2B and Appendix A). Our results confirm that the hiPSMEN1-MNA-6 cell line derived from the patient with MEN1 syndrome displays a differentiation potential comparable to that of the gold-standard H9 cell line.

Therefore, we demonstrated the high capacity of the established hiPSMEN1-MNA-6 cell line to directly differentiate into the endodermal lineage as a necessary step for the further creation of an isogenic model of MEN1 syndrome.

### 2.3. CRISPR/Cas9-Mediated Correction of the MEN1 Gene in Patient-Derived iPSCs

To correct disease mutation in the patient-derived iPSCs and generate isogenic cell lines, we employed CRISPR/Cas9 technology. We designed single-guide RNAs (sgRNAs) targeted to a genomic sequence of interest for Cas9-mediated allele-specific editing. Using online tools, we ranked sgRNAs according to their on-target specificity and the number of predicted off-target interactions. The final 20-nucleotide (nt) sgRNA was defined considering its minimum off-target activity and relatively high efficiency (Figure 3A, Supplementary sequences). For precise genome editing by homology-directed repair (HDR), the distance of the mutation from the CRISPR/Cas9 cut site was taken into account. As HDR-mediated incorporation of intended sequence change occurs most efficiently when the mutation is closest to the cut site [19], we chose the sgRNA with an optimal distance of 1 bp.

To further improve HDR efficiency, we used a rational design for a single-stranded oligodeoxynucleotide (ssODN) serving as a DNA repair template [20]. As a result, a 127 nt long asymmetric ssODN, complementary to the non-target strand and overlapping the Cas9 cut site with 36 bp on the 5′-end of the non-target strand, was generated (Supplementary sequences). To increase HDR accuracy, we included a silent Cas9-blocking mutation in the donor ssODN within the region corresponding to the sgRNA seed sequence. Such silent mutations can be safely introduced into the coding regions of edited genes and prevent further targeting and re-cutting of the corrected locus [21]. To facilitate clonal screening after genome editing, an artificial *HhaI* restriction site was added to the ssODN sequence (Figure 3A).

We used all-in-one plasmid delivery for the CRISPR/Cas9 components. The selected sgRNA was cloned into a vector expressing Cas9 nuclease linked by 2A peptide with an orange fluorescent protein (OFP) reporter gene allowing for fluorescence-activated cellsorting (FACS) analysis and monitoring of the transfection efficiency. The Cas9/gRNA co-expression plasmid was co-delivered with the ssODN donor into hiPSMEN1-MNA-6 cells via lipofection. We used a single-cell suspension reverse transfection protocol, which greatly increased gene transfer to hiPSCs, albeit with higher cell death rates. To retain cell viability during clonal isolation, FACS analysis was performed 48 h after transfection.

We collected OFP-expressing hiPSMEN1-MNA-6 cells in single-cell mode in a 96-well plate format. In order to define the optimal conditions for single-cell cloning, we plated post-FACS cells in two different manners. Firstly, we sorted OFP-positive hiPSMEN1-MNA-6 cells onto a feeder layer of patient-derived fibroblasts in a chemically defined mTeSR1 medium supplemented with KnockOut Serum Replacement and Rho-associated kinase inhibitor. Another approach was carried out in feeder-free culture using CloneR serum-free supplement and iPSC-conditioned medium. Both protocols led to a marked improvement in cloning efficiency, whereas the iPSC survival following flow sorting in standard culture medium without feeder or conditioned media was close to zero (data not shown). It should be noted that the CloneR/conditioned-medium-based protocol resulted in a 1.3-fold increase in colony formation efficiency compared to the ROCK inhibitor/feeder-based one.

Finally, more than 100 edited single colonies were picked and expanded for genomic DNA isolation and further analysis. A screening of the first 25 edited hiPSMEN1-MNA-6 cell clones using *HhaI* restriction digest, followed by sequencing, revealed four clones with desired modifications to the mutated *MEN1* allele (hiPSMEN1-MNA-6-e65, hiPSMEN1-MNA-6-e67, hiPSMEN1-MNA-6-e85, and hiPSMEN1-MNA-6-e87). Along with the correction of the disease mutation (TAC→GAC at codon 418), we successfully detected the silent Cas9-blocking mutation on the edited DNA strand, indicating that the designed ssODN was an efficient HDR donor (Figure 3B). It should be noted that there were no non-homologous end-joining (NHEJ) events associated with HDR in any of the four clones, suggesting that the wild-type *MEN1* allele remained unaffected. Moreover, 16% HDR efficiency was achieved (Figure 3C) without any pretreatment of cells, which is much higher than the rates that are usually reported for pluripotent cell lines [22].

### 2.4. Validation of the Corrected Isogenic iPSC Lines

To ensure the specificity of the sgRNA/Cas9 system, we examined in silico predicted off-target sites in the genomes of the edited iPSC clones using Sanger sequencing. We applied a web-based software tool developed to rank the cutting frequencies among genome-wide targets [23]. According to the highest score, we chose four potential off-target regions located on chr11:64572603-64572622, chr11:7466134..7466153, chr10:76789631..76789650, and chr2:111664534..111664553 (Appendix A). As expected, the first one corresponds to the wild-type MEN1 locus. To exclude non-specific cutting of the wild-type allele, we analyzed hiPSMEN1-MNA-6 clones with NHEJ events by matching both DNA strands. Sequence analysis revealed small deletions (3–12 bp) near the predicted Cas9 cut site in one allele, while no disruption was observed in the other allele.

It seems important to highlight that the sgRNA design must be stricter and more accurate for allele-specific genome editing. The proximity of the mutated nucleotide to the PAM sequence, as close as possible to the 3′ end of the seed sequence of the sgRNA, should be taken into account [24]. We relied on this strategy and, as a result, observed no Cas9-mediated cleavage of the wild-type MEN1 allele among the screened edited clones. As for other off-target sites, we assessed their effects in vitro in the correctly modified hiPSMEN1-MNA-6 cell clones. Sanger sequencing of the corresponding PCR products demonstrated intact genomic DNA with no indels or point mutations. The sequences flanking each of the three off-target sites were identical in all clones (Figure 4A).

Importantly, the iPSC clones with the desired correction in the MEN1 gene maintained their pluripotency after genome editing. The ESC-like morphology of the edited cells was observed during colony expansion and further serial passaging (Figure 4B). The immunostaining against pluripotency markers, including OCT4, SOX2, SSEA-4 (Figure 4C), and TRA-1-60 (Appendix A) also confirmed the pluripotent state of the generated isogenic cell lines. In addition, the isogenic iPSCs exhibited the ability to differentiate into all three germ layers (Figure 4D). Finally, all the edited clones (hiPSMEN1-MNA-6-e65, hiPSMEN1-MNA-6-e67, hiPSMEN1-MNA-6-e85, and hiPSMEN1-MNA-6-e87) were authenticated via STR profiling. The analysis revealed matching of 20 loci when compared to the parental hiPSMEN1-MNA-6 cell line (Appendix A).

In summary, we generated a patient-specific iPSC line carrying the MEN1 mutation and modified the target allele through CRISPR/Cas9 editing with great specificity and efficiency. The high potential of these cells to differentiate into the endodermal lineage ensures the next steps in the development of more specialized cells, which are commonly affected in MEN1 patients, such as parathyroid or pancreatic islet cells. This isogenic system will serve as a novel, powerful in vitro platform for studying MEN1 syndrome.

## 3. Discussion

Various animal models of MEN1 syndrome have been generated and characterized. In contrast to the MEN1-null drosophila model, homozygous *Men1* knockout mice died at embryonic stages E11.5–13.5 [4,25]. Whole-body and tissue-specific heterozygous *Men1* knockout mice have been widely used to unravel the function of menin in endocrine tumorigenesis. Most of these studies considered the central role of menin as a tumor-suppressor [26,27,28]. However, animal models are not able to completely elucidate the exact molecular pathways involved in MEN1-associated tumor development and progression. First and foremost, *Men1*-deficient mouse models can only partially recapitulate the pathological processes and features of the disease due to their species-specific genetic background. Moreover, such models are generated by large gene deletions that were not observed in patients. This leads to differences in tumor frequency and spectrum between mice and humans that hamper basic research and preclinical testing [6]. This is especially crucial regarding non-functioning pancreatic neuroendocrine tumors, which are not developed in *Men1* knockout mice but serve as the main cause of MEN1-related mortality in patients [29,30].

Model systems based on human cells would provide a deeper insight into the molecular mechanisms of MEN1 pathology. Nevertheless, the currently available in vitro models are restricted to either non-endocrine cell types, such as 293T [31], or neuroendocrine cell lines BON-1 and QGP-1 not harboring mutations in the *MEN1* gene [32]. Hence, questions have been raised regarding their relevance. There is an urgent need for an appropriate cell model of MEN1-associated tumorigenesis that will enable understanding of the underlying biology and the identification of potential molecular therapeutic targets in a tissue-specific context.

Patient-derived iPSCs provide a unique and powerful opportunity to address these issues. Such iPSCs retain a specific genetic background and are capable of differentiation into multiple somatic tissues affected by hereditary diseases. This overcomes the lack of renewable sources of the patient’s primary cells [33]. However, one crucial challenge of iPSC-based technology is the genetic and epigenetic variability among individual cell lines, which can impact their differentiation potential and thus form an obstacle to comparisons of normal and disease phenotypes [34]. The generation of genetically matched iPSC lines, with the sole exception of disease mutation, allows for an investigation into the disease’s pathology in completely defined isogenic conditions.

Recent achievements in genome editing technologies enabling site-specific correction of patient iPSCs—particularly the rapid progress in CRISPR/Cas9 systems—substantially facilitate isogenic disease modeling [33,34,35]. Although this approach has been successfully applied for different disorders, including neurological disorders [36] and several other monogenic diseases [37,38,39], there is a lack of such isogenic models for hereditary cancer. Hadoux et al. generated a patient-derived iPSCs carrying the *RET* mutation along with the CRISPR/Cas9-corrected isogenic control to unravel the molecular mechanisms underlying the multiple endocrine neoplasia type 2A [40]. However, there have been no isogenic models of MEN1 syndrome to date. The present work is the first study, to the best of our knowledge, to report a patient-specific isogenic iPSC-based system enabling the investigation of the pathogenesis of MEN1 syndrome.

Herein, we established a new patient-specific iPSC line carrying a heterozygous *MEN1* mutation (c.1252G>T, D418Y) and performed allele-specific CRISPR/Cas9 genome editing. Strikingly, only one out of four successfully edited clones contained the artificial *HhaI* restriction site. We suppose that such an atypical repair outcome might occur due to the multi-stage repair pathway [41]. After CRISPR/Cas9-mediated cleavage, only the right homology arm of ssODN anneals with the DNA strand upstream of the predicted cut site, while complementary DNA from the homologous chromosome can be utilized as a template to repair the break and further DNA synthesis and elongation. As a result, we observed an imprecise repair outcome downstream from the cut site. Importantly, the generated iPSCs showed endodermal differentiation potential similar to that of the gold-standard ESCs. Our results prove that the generated isogenic pair of cell lines can be further differentiated into parathyroid or pancreatic lineages and comprehensively used for functional studies. This differentiation capacity is of the greatest interest for considering the clinical picture of MEN1 patients suffering from multiple gland parathyroid hyperplasia, along with non-functioning pancreatic neuroendocrine tumors [13].

Although Guo et al. previously reported an iPSC line with artificially introduced *MEN1* mutation [42], their model cannot take into account the genetic variations between original wild-type and patient-derived cells, which might confuse further analysis. Only a genetically defined cell model, such as we have described here, allows for reliable identification of the true disease-relevant genes and phenotypes.

It is noteworthy that the mutation at codon 418 is relatively common, observed many times in unrelated families with MEN1 syndrome. Substitutions of aspartic acid by histidine (GAC→CAC) [43], asparagine (GAC→AAC) [44], and tyrosine (GAC→TAC) [2,45] have been previously reported as pathogenic variants. According to the Universal Mutation Database (URL accessed on 5 September 2021 at: http://www.umd.be/MEN1/), the D418N mutation is one of the most frequent of the causal variants identified in MEN1 syndrome. Previously, Yaguchi et al. suggested that menin with missense D418N mutation might be degraded through the ubiquitin-proteasome pathway [46], based on the measurements of the exogenous protein level in the HEK293 cells. We did not detect any alteration in menin expression level or cellular localization in the patient-derived iPSCs carrying heterozygous D418N mutation. Menin expression was comparable to that in the normal ESC line. Our data are consistent with a report by Wautot et al., showing that endogenously expressed menin did not differ between cell lines derived from MEN1 patients and healthy donors [47], which supposes the compensation mechanism by wild-type allele upregulation.

In conclusion, we developed a patient-specific isogenic model for MEN1 syndrome, applying cellular reprogramming and CRISPR/Cas9 genome editing. We expect that the established isogenic system of iPSC lines, which differ by only a single mutation in residue D418 of menin, can help us to determine the still unknown factors underlying MEN1 pathology and will be a valuable in vitro tool for the research community.

## 4. Materials and Methods

### 4.1. Ethical Statement

This study was approved by the Ethics Committee of the Endocrinology Research Centre (Moscow, Russia), protocol No. 12/19.08.2020. The skin tissue sample was obtained with the patient’s written informed consent.

### 4.2. Isolation and Maintenance of Dermal Fibroblasts

Dermal fibroblasts were obtained from a patient’s skin biopsy sample (~1.5 mm^2^). The skin was dissected into small pieces and incubated in culture medium containing DMEM (Gibco, Waltham, MA, USA) with 10% fetal bovine serum (FBS) (HyClone, Marlborough, MA, USA), 0.1 µM non-essential amino acids (Paneco, Moscow, Russia), 1% Glutamax (Gibco), and 1% penicillin/streptomycin (Gibco) in a 6-well plate coated with glass coverslips, at 37 °C and 5% CO_2_. When cells started to sprout from the explants, the culture medium was changed every 3 days. After about 2 weeks, a confluent fibroblast monolayer was observed. Confluent cultures were passaged using 0.25% trypsin (Gibco). Fibroblasts were cryopreserved in DMEM with 20% FBS and 10% DMSO (Sigma, St. Louis, MO, USA) at −150 °C. The absence of mycoplasma contamination was confirmed by PCR analysis using a MycoReport Kit (Evrogen, Moscow, Russia).

### 4.3. iPSC Generation

Patient-derived fibroblasts were reprogrammed into iPSCs using a ReproRNA-OKSGM kit (STEMCELL Technologies, Vancouver, BC, Canada) according to the manufacturer’s instructions. Before transfection, puromycin titration (Sigma) was performed to define the optimal concentration (3 µg/mL) for antibiotic selection. For reprogramming, 1 × 10^5^ fibroblasts were seeded on a matrigel-coated 35 mm dish and then transfected with ReproRNA-OKSGM vector followed by puromycin selection. After 14–21 days, a number of ESC-like colonies formed from successfully transfected fibroblasts. Colonies were manually picked and expanded in mTeSR™1 medium (STEMCELL Technologies) on Matrigel hESC-qualified matrix (Corning, New York, NY, USA).

### 4.4. iPSC and ESC Cultures

Human embryonic stem cell lines H9, hESKM-05 [48], and iPS12 [49] were used as control cell lines. The hiPSMEN1-MNA-6, edited clones, and control ESCs were maintained in feeder-free conditions (mTeSR™1 medium, Matrigel hESC-qualified matrix) with daily medium change. For passaging, cells were treated with Gentle Cell Dissociation Reagent (Gibco) and replated on Matrigel hESC-qualified matrix in mTeSR™1 medium containing 10 µM Rho-associated kinase (ROCK) inhibitor Y-27632 (STEMCELL Technologies). Cells were cryopreserved in a freezing medium (10% DMSO in FBS) at −150 °C. All cell lines tested negative for mycoplasma contamination.

### 4.5. Immunofluorescence Analysis

Cells were fixed with 4% paraformaldehyde for 15 min at room temperature (RT), then washed with phosphate-buffered saline (PBS) and permeabilized with 0.2% Triton X-100 in PBS for 10 min. After washing, cells were incubated in a blocking solution (2.5% BSA, 0.1% Tween 20 in PBS) for 30 min at RT. Following overnight incubation with primary antibodies at 4 °C, cells were washed and subjected to incubation with appropriate secondary antibodies for 1 h at RT in the dark. Antibodies and final dilutions are given in Appendix A. Primary antibodies directly conjugated with fluorophores were used according to the manufacturer’s instructions. DAPI was used for nuclear counterstaining for 10 min at RT in the dark. Fluorescence visualization was performed using a Leica DMi8 microscope (Leica Microsystems, Wetzlar, Germany), CELENA S Digital Imaging System (Logos Biosystems, Gyunggi-do, South Korea), or Cytation 3 Cell Imaging Multi-Mode Reader (BioTek, Winooski, VT, USA).

### 4.6. Karyotyping

At passage 22, iPSCs grown in the logarithmic phase were treated with Colcemid (Gibco) for 1 h. Cells were then trypsinized, treated with a hypotonic solution for 15 min, and fixed in methanol–acetic acid. Metaphases were spread on microscopic slides, and standard G-banding analysis of at least 10 metaphase spreads was performed.

### 4.7. In Vitro Differentiation into Three Germ Layers

The iPSCs were differentiated through embryoid body (EB) formation. Cells grown to 90% confluency were treated with Gentle Cell Dissociation Reagent (Gibco) and plated in an ultra-low-attachment 24-well plate (Corning) in Essential 8 medium (Gibco) with 10 µM ROCK inhibitor. After 24 h of incubation at 37 °C in 5% CO_2_, the cells formed EBs, which were gently collected with a 2 mL serological pipette and plated in an untreated 6-well culture plate (Eppendorf, Enfield, CT, USA) in DMEM/F12 (Gibco) supplemented with 20% KnockOut Serum Replacement (Gibco), 1% Glutamax, 0.1 µM non-essential amino acids, and 0.1 µM β-mercaptoethanol (Sigma). To induce spontaneous trilineage differentiation, the medium was gradually substituted with DMEM with 10% FBS, 1% Glutamax, and 0.1 µM non-essential amino acids (Paneco) during the first week of cultivation. This medium was then changed every 3 days for the next two weeks. On Day 21, the differentiated cells were analyzed by immunostaining using antibodies against markers for the endoderm, mesoderm, and ectoderm. The antibodies and final dilutions are shown in Appendix A.

### 4.8. Directed Differentiation into the Definitive Endoderm

H9 cells at passage 57 and patient-derived hiPSMEN1-MNA-6 cells at passage 20 were differentiated into the endodermal lineage using a Stemdiff Definitive Endoderm Kit (STEMCELL Technologies) according to the manufacturer’s instructions, with the exception of initial cell density. Differentiation was induced on Day 1 when cells were approximately 30–35% confluent. On Day 5, qualitative and quantitative assays were performed via immunocytochemistry and flow cytometry, respectively. The antibodies used are listed in Appendix A.

### 4.9. Flow Cytometry

On the fifth day of DE differentiation, monolayer cell cultures were dissociated to single-cell suspensions with Versene solution (Paneco) for 10 min at 37 °C in 5% CO_2_ and then collected by centrifugation at 300× *g* for 5 min. Cells were fixed with 2% paraformaldehyde for 5 min at room temperature, then centrifuged at 600× *g* for 5 min and permeabilized in 0.2% Triton X-100 in PBS for 5 min. After 30 min of incubation in blocking solution (2.5% BSA, 0.1% Tween 20 in PBS), primary antibodies were added for 3 h at RT. Cells were washed once with 0.1% Tween20 in PBS and subjected to incubation with appropriate secondary antibodies conjugated to FITC for 1 h at RT in the dark. Antibodies and final dilutions are given in Appendix A. Flow cytometry analysis was performed using a NovoCyte Flow Cytometer (ACEA Biosciences, San Diego, CA, USA). Positive cells were gated on a dot plot of FSC versus FITC fluorescence.

### 4.10. sgRNA Design and Cloning

The Benchling CRISPR (URL accessed on 7 September 2020 at: https://benchling.com/) and CHOPCHOP (v.3) (URL accessed on 7 September 2020 at: https://chopchop.cbu.uib.no/) online tools were used to design sgRNA and evaluate its off-target effects. The chosen sgRNA sequences were cloned into the GeneArt CRISPR Nuclease Vector (Thermo Fisher Scientific, Cleveland, OH, USA) under the control of a U6 promoter. Five extra bases were added to the 3′ end of each single-strand sgRNA oligonucleotide to enable direct cloning into the vector. The corresponding single-strand oligonucleotides were synthesized by Evrogen. The oligonucleotide sequences are given in Appendix A. The oligonucleotides were annealed and ligated with the GeneArt CRISPR Nuclease Vector. Several ampicillin-resistant colonies obtained after transformation of E. coli were analyzed via plasmid DNA isolation. To verify correct sgRNA insertion into the GeneArt CRISPR Nuclease Vector, plasmid DNA was sequenced using the U6 promoter forward primer 5′-GGACTATCATATGCTTACCG-3′. The GeneArt-sgRNA-MEN1 plasmid DNA used for iPSC transfection was purified using an EndoFree Plasmid Maxi kit (Qiagen, Germantown, MD, USA).

### 4.11. iPSC Transfection

Patient-derived iPSCs were transfected by reverse lipofection with TransIT^®^-LT1 Transfection Reagent (Mirus Bio, Madison, WI, USA). The procedure was performed in 35 mm plate-format according to the manufacturer’s instructions. For the transfection, 1.4 × 10^6^ cells were used. The transfection mix contained 12 μL of TransIT^®^-LT1 Transfection Reagent, 4 μg of GeneArt-sgRNA-MEN1 plasmid DNA, and 0.8 μL of ssODN 100 μM stock solution. The transfection efficiency was monitored by determining OFP expression via fluorescence microscopy using a CELENA S Digital Imaging System (Logos Biosystems).

### 4.12. Clone Selection

At 48 h post-transfection, OFP-expressing hiPSMEN1-MNA-6 cells were analyzed and collected by fluorescence-activated cell sorting (FACS). In brief, cells were treated with EDTA solution, resuspended in mTeSR1 medium, and filtered through a 70 μm cell strainer. Next, cells were loaded into a Sony MA900 sorter (Sony Biotechnology Inc., Tokyo, Japan). Detection of the target population of OFP-positive cells was performed using a 488 nm laser in both the PE and FITC channels to separate the fluorescence signal from the autofluorescence background. Cells were sorted in single-cell mode in 96-well plates. The culture medium was changed every day. Discernible iPSC colonies were observed after one week of incubation. Selected hiPSMEN1-MNA-6 edited clones were then expanded for further screening and cryopreservation.

### 4.13. DNA Sequencing

Genomic DNA was isolated from the monolayer cell culture using a commercial column-based DNA extraction kit (ExtractDNA Blood kit, Evrogen). To confirm the presence of c.1252G>T mutation in patient-derived fibroblasts and iPSCs, or to analyze DNA changes in CRISPR/Cas9-edited iPSC clones, the target region of exon 9 in the MEN1 gene was amplified by PCR from genomic DNA using a pair of primers (forward 5′-TACGGGATTAGGGATGGCAG-3′, reverse 5′-GGGCCAGAAAAGTCTGACAA-3′). The primers used to examine off-target effects in CRISPR/Cas9-edited iPSC clones are shown in Appendix A. The PCR products were purified using a Cleanup Standard kit (Evrogen) followed by Sanger sequencing (Evrogen).

### 4.14. STR Analysis

STR (short tandem repeat) analysis was performed on hiPSMEN1-MNA-6 cells and edited cell clones. Genomic DNA from monolayer cultures was extracted using an ExtractDNA Blood Kit (Evrogen). In total, 20 distinct polymorphic loci were characterized using a COrDIS Plus STR Amplification Kit (Gordiz, Moscow, Russia).

## Figures and Tables

**Figure 1 ijms-22-12054-f001:**
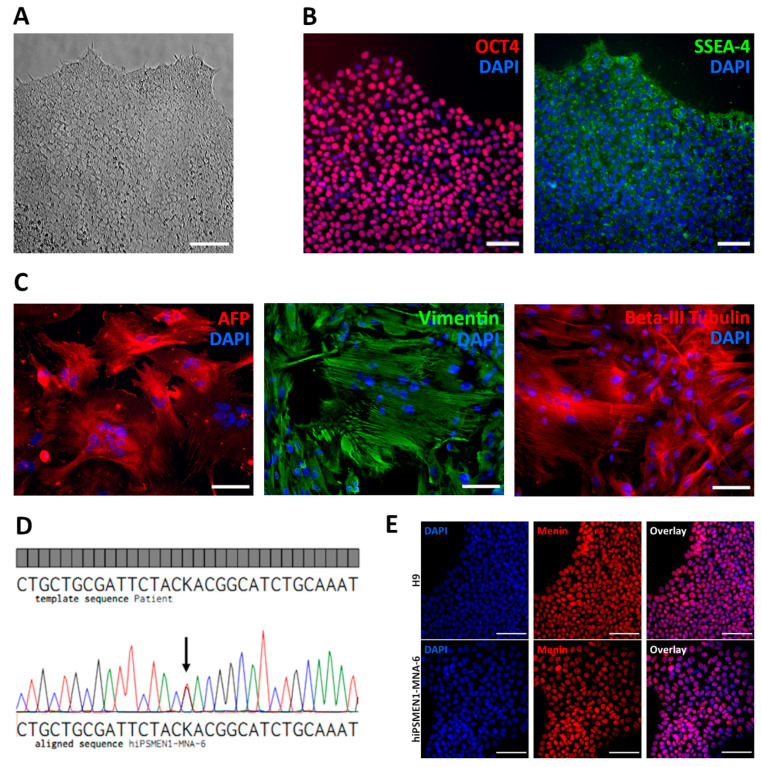
Characterization of hiPSMEN1-MNA-6 cell line derived from a patient with MEN1 syndrome. (**A**) Brightfield microscopy image of iPSC colony showing a typical human embryonic stem-cell-like morphology. Scale bar represents 100 µm; (**B**) Immunostaining for transcription factor OCT4 (red) and surface antigen SSEA-4 (green) in hiPSMEN1-MNA-6 cells, showing the expression of pluripotent genes. Nuclei were counterstained with DAPI (blue). Scale bars represent 100 µm; (**C**) Differentiation of hiPSMEN1-MNA-6 cells into all three germ layers (endoderm, mesoderm, ectoderm). Immunostaining for Alpha-fetoprotein, AFP (red), Vimentin (green) and Beta-III-tubulin (red), demonstrating the expression of endodermal, mesodermal and ectodermal markers, respectively. Nuclei were counterstained with DAPI (blue). Scale bars represent 100 µm; (**D**) Sanger sequencing of the region of interest in the *MEN1* gene in patient’s sample (top) and hiPSMEN1-MNA-6 cells (bottom). The analysis revealed that cells sustained a heterozygous mutation c.1252G>T (indicated with a black arrow) in the *MEN1* gene; K = T or G according to the IUPAC nucleotide code; (**E**) Immunofluorescence analysis of expression of menin (red) in H9 (upper panel) and hiPSMEN1-MNA-6 (lower panel) cell lines. Nuclei were counterstained with DAPI (blue). Scale bars represent 100 µm.

**Figure 2 ijms-22-12054-f002:**
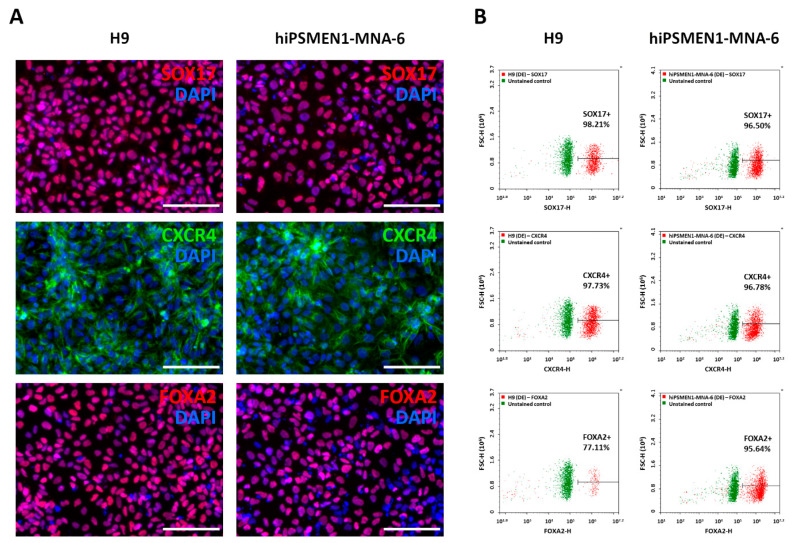
Assessment of the definitive endoderm (DE) differentiation potential of the established patient-derived hiPSMEN1-MNA-6 cells in comparison to the human embryonic stem cell line H9 (gold-standard). (**A**) Immunostaining for DE markers SOX17 (red), CXCR4 (green), and FOXA2 (red). DAPI was used to counterstain cell nuclei (blue). Scale bars represent 100 µm; (**B**) Flow cytometry analysis for DE markers SOX17, CXCR4, and FOXA2. Data are presented as dot plots of forward scatter (FSC) versus DE marker fluorescence (in the FITC fluorescence channel). The horizontal marker was set based on background fluorescence from unstained cells. Populations of SOX17+, CXCR4+, and FOXA2+ cells and unstained cells are shown in red and green, respectively.

**Figure 3 ijms-22-12054-f003:**
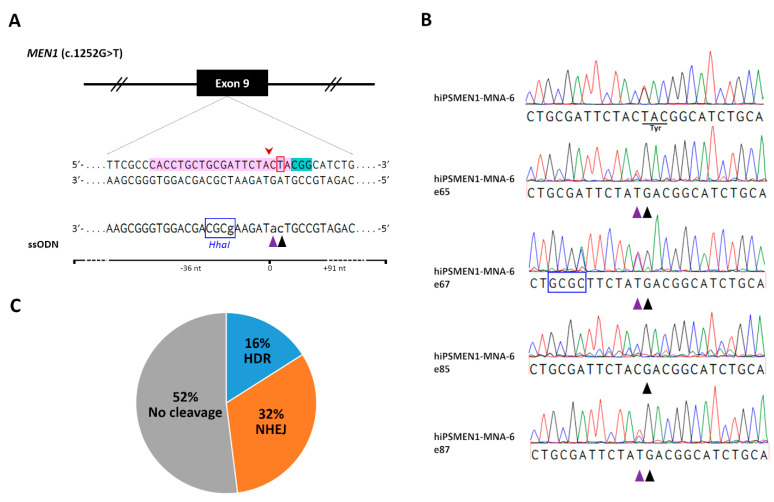
CRISPR/Cas9-mediated correction of the c.1252G>T in the *MEN1* gene in patient-specific iPSCs. (**A**) Schematic representation of the mutated region of *MEN1* with the corresponding ssODN repair template. The magnified view illustrates the sgRNA target site (shown in magenta) and the PAM sequence (shown in cyan) in the *MEN1* gene. The mutated nucleotide (thymidine) is labeled with a red open box. The potential Cas9 cleavage site is indicated by a red arrowhead. Three intended point mutations in the ssODN are indicated by lowercase letters, where Cas9-blocking mutation and the desired mutation (cytosine) are indicated with a purple and a black triangle, respectively. The *HhaI* restriction site is labeled with a blue open box; (**B**) Sequencing chromatograms of patient-specific hiPSMEN1-MNA-6 cells harboring heterozygous mutated alleles of *MEN1* and successfully edited clones (e65, e67, e85, e87) with wild-type repaired alleles. The potential Cas9 cleavage site is indicated by a red arrowhead. The tyrosine (mutated) and the aspartic acid (wild-type) codons are underlined. The triangles indicate Cas9-blocking mutation (purple) and the desired nucleotide change (black). The *HhaI* restriction site is labeled with a blue open box. Clone hiPSMEN1-MNA-6-e67 contains all three desired mutations; (**C**) A pie chart shows the pattern of CRISPR-Cas9-induced modifications of *MEN1* in patient-specific iPSCs. HDR—precision editing by homology-directed repair, NHEJ—deletions caused by non-homologous end joining.

**Figure 4 ijms-22-12054-f004:**
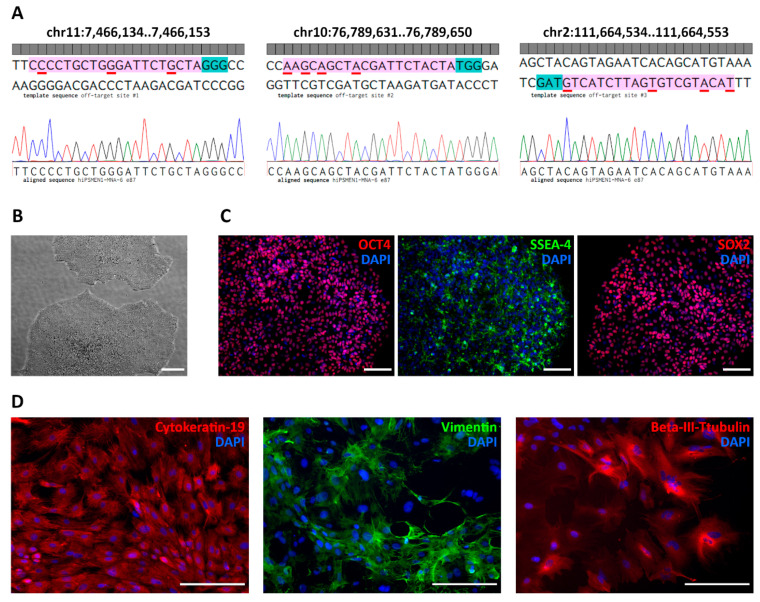
Characterization of gene-corrected isogenic control iPSC line hiPSMEN1-MNA-6 e87 derived from a patient with MEN1 syndrome. (**A**) Analysis of putative genomic off-target loci. Top: plus and minus DNA strands, where the red underlines indicate mismatches; 20-base regions for sgRNA off-target site and PAM sequences are shown in magenta and cyan, respectively. Bottom: sequencing chromatogram of the corresponding region in the genome of hiPSMEN1-MNA-6 e87 cells; (**B**) Representative appearance of the colonies demonstrating a typical human embryonic stem cell-like morphology in hiPSMEN1-MNA-6 e87 culture. Brightfield microscopy image, scale bar represents 100 µm; (**C**) Immunostaining for pluripotent markers OCT4 (red), SSEA-4 (green) and SOX2 (red) in hiPSMEN1-MNA-6 e87 cells confirming that the pluripotency was maintained. DAPI was used to counterstain cell nuclei (blue). Scale bars represent 100 µm; (**D**) Differentiation of hiPSMEN1-MNA-6 e87 cells into all three germ layers (endoderm, mesoderm, ectoderm). Immunostaining for Cytokeratin-19 (red), Vimentin (green) and Beta-III-tubulin (red), demonstrating the expression of endodermal, mesodermal and ectodermal markers, respectively. Nuclei were counterstained with DAPI (blue). Scale bars represent 200 µm.

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
