# Peer review of "A Novel Isogenic Human Cell-Based System for MEN1 Syndrome Generated by CRISPR/Cas9 Genome Editing"

_ijms, 2021, doi:10.3390/ijms222112054_

Round 1

Reviewer 1 Report

In this manuscript, Klementieva and colleagues report the creation of iPSC cell line from a patient containing a missense heterozygous mutation in exon 9 of the MEN1 gene. The authors then corrected the mutation with HDR using CRISPR/Cas9 and ssODNs. Next they show that these gene corrected iPSCs retained pluripotency after editing. The study is interesting but several experiments and controls are missing. I have suggested major revisions and additional experiments as listed below:

Major comments

-Conceptually, it is difficult to understand what authors mean by “reduce the genetically determined variability of the cellular models”. MEN1 mutations are variable and thus there is bound to be genetically different cellular models especially if the aim to correct the underlying mutation with gene editing. Please clarify or replace.

-Figure 1: A very important control is missing. The authors need to use a normal iPSC or the H9 line for a side-by-side comparison of differentiation and expression of different markers. At the moment, the images are only qualitative.

-Figure 2:

(a) Please show control/non-stained cells (at least in the supplementary).

(b) How many times was the flow cytometry experiment performed? As it is, the plots don’t look convincing. A histogram plot in the supplementary is needed for clarify. Please include unstained controls that were used to set the gates. A bar graph comparing SOX17, CXCR4 and FOXA2 expression in H9 vs hiPSMEN1-MNA-6 is absolutely essential. So is a statistical analysis to test differences between the two groups. As it is, there seems to be a significant difference in the expression of FOXA2 in H9 vs hiPSMEN1-MNA-6 which does not support authors’ conclusion.

-Figure 3:

Please comment why all only one out of four clones gained the HhaI restriction site. If the cells used the supplied ssODNs as a template for correction, this proportion should have been higher. Please provide explanations.

-Figure 4:

The authors need to compare hiPSMEN1-MNA-6 and hiPSMEN1-MNA-6 e87 for their ability to differentiate into the three germ layers. TRA-1-60 is mentioned in the text but not provided in the figure.

-Finally, the authors claim that this model can be used to study the underlying biology of MEN1 mutations. However, no examples with other similar studies have been discussed. The authors should also discussion or suggest how this could be achieved.

Author Response

We thank the Reviewer for the careful and insightful reading of our manuscript. We appreciate all the suggestions, which have been very helpful in improving the manuscript. Herein we submit the revised version of the manuscript addressing the Reviewer’s comments and present our responses to each of them.

Reviewer 2 Report

Dear Editor, Dear Authors,

Thank you very much for considering my expertise in the revision process of the article; “A novel isogenic human cell-based system for MEN1 syndrome generated by CRISPR/Cas9 genome editing”, by Klementieva et al., submitted for publication in the International Journal of Molecular Sciences.

In the following article authors have aimed to develop a new human cell-based model, for investigating the multiple endocrine neoplasia type 1 (MEN1) syndrome. MEN1 is a rare tumor syndrome, characterized by its complex manifestation and no obvious phenotype-genotype correlations. By using CRISPR/Cas9 technology authors have corrected the missense heterozygous mutation (c.125G>T) of the dermal primary fibroblast cell culture, isolated from a skin biopsy of a 39-year old female patient. Moreover, they have reprogrammed the fibroblasts cells into pluripotent cells by non-targeting RNA based system. According to the author the high potential of these cells to get differentiated into endodermal lineage ensures the further development of more specialized cell types that are commonly affected in MEN1 patient.

The current study submitted for publication in the Journal is accurately accomplished, with a clear-cut conclusions and flawless experimentally supported results. The experiments are extremely well documented and the overall experimental strategy may serve as a guideline to establish similar model systems for variety of known mutations associated with tumor development. However, despite the superior impression there are few points, which needs to be addressed before the paper is considered for publication in the Journal.

  1. Authors have provided a comprehensive genetic characterization of the MEN1 mutated cell line. However, there is no single experiment showing the expression level of the menin as well as its level after correction. It is documented that at least D418N mutation results in low protein level of MEN1 (DOI: 10.1128/MCB.24.15.6569–6580.2004) and it will be great is authors show this in their model system.
  2. The chromosomes in the karyotype on Figure 1D are quite small to be informative. It will be nice if the image size is increased even if it is necessary to put the image as a supplementary figure.

Author Response

(The authors gave the same response as above.)
